# Cross-context News Corpus for Protest Events related Knowledge Base Construction

**Ali Hürriyetoğlu**                                                         AHURRIYETOGLU@KU.EDU.TR
**Erdem Yörük**                                                                 ERYORUK@KU.EDU.TR
**Deniz Yüret**                                                                 DYURET@KU.EDU.TR
**Osman Mutlu**                                                                 OMUTLU@KU.EDU.TR
**Çağrı Yoltar**                                                                CYOLTAR@KU.EDU.TR
**Fırat Duruşan**                                                              FDURUSAN@KU.EDU.TR
**Burak Gürel**                                                                 BGUREL@KU.EDU.TR
*Rumelifeneri Yolu 34450*
*Sarıyer, İstanbul / Türkiye*

## Abstract

We describe a gold standard corpus of protest events that comprise of various local and international sources from various countries in English. The corpus contains document, sentence, and token level annotations. This corpus facilitates creating machine learning models that automatically classify news articles and extract protest event-related information, constructing knowledge bases which enable comparative social and political science studies. For each news source, the annotation starts on random samples of news articles and continues with samples that are drawn using active learning. Each batch of samples was annotated by two social and political scientists, adjudicated by an annotation supervisor, and was improved by identifying annotation errors semi-automatically. We found that the corpus has the variety and quality to develop and benchmark text classification and event extraction systems in a cross-context setting, which contributes to the generalizability and robustness of automated text processing systems. This corpus and the reported results will set the currently lacking common ground in automated protest event collection studies.

## 1. Introduction

Socio-political event knowledge bases enable comparative social and political studies. The spatio-temporal distribution of these events sheds light on the causes and effects of governmental service management and political discourse resonate in the society. Consequently, construction of these knowledge bases a relatively long history. We focus on the protest events as a type of socio-political events. These events are in the scope of contentious politics (CP) and characterized by riots, strikes, and social movements, i.e. the "repertoire of contention" [Giugni, 1998, Tarrow, 1994].

Continuity of news articles and the significant impact of socio-political events direct social and political scientists to exploit news data to create knowledge bases of these events [Chenoweth and Lewis, 2013, Weidmann and Rød, 2019, Raleigh et al., 2010]. The need for collecting socio-political event data, which is protest and conflict data in the context of our work, has been satisfied manually [Yoruk, 2012], semi-automatically [Nardulli et al., 2015], and automatically [Leetaru and Schrodt, 2013, Boschee et al., 2013, Schrodt et al., 2014, Sönmez et al., 2016]. Protest knowledge base creation is either too expensive

for manual approaches or has serious limitations when it is performed automatically [Wang et al., 2016, Ettinger et al., 2017, Ward et al., 2013]. Moreover, there has not been any common ground across projects that would enable to compare results of these studies [Lorenzini et al., 2016]. Therefore, as Emerging Welfare (EMW) project, we took on the challenge to create the common basis in terms of required high quality data and state-of-the-art tools for fully automating the creation of reliable and valid protest knowledge bases in a way that would serve as a benchmark and enable protest event collection studies benefit from it. This effort yielded a gold standard corpus (GSC) that will serve the machine learning and computational linguistics communities to study text processing tool development for constructing knowledge bases of protest events as well.

The GSC consists of English news articles from various local and international sources [1] from India, China, and South Africa. Variety of the sources has allowed studying cross-context robustness and generalizability, therefore addressing style and content change across sources, which are critical requirements of the ML models. The annotations were applied at document, sentence, and token level subsequently. Each instance is annotated by two people and adjudicated by the annotation supervisor for each level. Moreover, a detailed manual and semi-automatic quality check and error analysis were applied on the annotations.

The corpus contains more than 10,000 news articles labelled as protest and non-protest; protest containing articles, which are more than 800, were further labelled as containing event information or not at the sentence level, and annotated at the token level for detailed event information such as the trigger(s), semantic category, place, time and actor(s) of events, and co-reference information. The corpus has enabled the development of a pipeline of machine learning (ML) models that extract protest events from an archive of news articles. Moreover, some parts of the corpus has enabled shared-tasks that validated the corpus for cross-context document and sentence classification and token extraction [Hürriyetoğlu et al., 2019a,b][2] and event sentence coreference identification [Hürriyetoğlu et al., 2020].

The contributions of this paper are 1) a robust methodology for creating a corpus that enables the creation of robust ML-based text processing tools, 2) insights from applying this methodology to news archives across contexts, 3) a corpus that contains data from multiple contexts and annotations at various levels of granularity, 4) results of a pipeline consisting of automated tools that are created using the corpus, and 5) first state-of-the-art recall quantification on real random data in contrast to recall measurement in limited settings.

We will describe the context of our work in reference to recent related work in Section 2. Next we will introduce our methodology, the manuals we prepared, and the corpus we have created in Sections 3, 4, and 5, respectively. Section 6 reports the results of the ML tools that were created using the corpus. Finally, Section 7 concludes this report by drawing overall results and point to future steps we are planning to perform in the near future.

## 2. Relevant work

Automated socio-political event collection is shaped around language resources; automated tools that exploit these resources; the assumptions made to complete the design of an event

---

1. International sources were filtered based on meta-information to focus on the case countries.

2. https://emw.ku.edu.tr/clef-protestnews-2019/ https://emw.ku.edu.tr/?event= challenges-and-opportunities-in-automated-coding-of-contentious-political-events

collection system; and datasets that are output of these systems. The shared resources are rare and the accessible tools are a few. Moreover, the assumptions made in delivering a resulted dataset are not examined in diverse settings. Automated tools for event information collection are designed in terms of pipelines that receive news articles from one or more news sources and yield records of event information. Each tool is inherently limited to the language resources it facilitates and the setting it is validated. Therefore, the quality of the result when an automated tool is used to analyze different sources, i.e. cross-context performance of these pipelines is rarely evaluated. One of the first steps of these pipelines, which is discriminating between relevant and irrelevant documents, has been extensively reported in Croicu and Weidmann [2015] and Hanna [2017]. Key term lists and labelled documents aid in determining which news reports contain relevant events. Each study reports their own key term lists and the way they use it. Moreover, labelled documents are released in terms of their URLs or document IDs in some collection without their content [Makarov et al., 2015]. Accessing the dataset using this limited information is the responsibility of the people who want to use these resources. Our work should be considered as a new version that does not restrict itself with keywords and apply state-of-the-art ML models to tackle the problem of selecting documents that contain event information, which is known as report selection problem in this field.

The second main step of this task is extracting relevant tokens. ACE [Doddington et al., 2004] and TAC-KBP [Mitamura et al., 2015] are language resources that can be exploitet for token level event extraction. The size and scope of the data for protest events provided by these resources is quite limited. Moreover, the event definition of ACE and TAC-KBP does not capture the CP events. For instance, the ACE definition of the event type DEMONSTRATE, in itself, is too restrictive to be applicable in terms of a broad understanding of CP for two reasons. First, as it seems to limit the scope of this event type to spontaneous (that is unorganized) gatherings of people, it excludes certain actions of political and/or grassroots organizations such as political parties and NGOs. Protest actions of such organizations sometimes do not involve mass participation despite aiming at challenging authorities, raising their political agendas or issuing certain demands. Putting up posters, distributing brochures, holding press declarations in public spaces are examples of such protest events. Secondly, the requirement of mass participation in a public area leaves many protest actions such as on-line mass petitions and boycotts, which are not necessarily tied to specific locations where people actually gather, and actions of individuals or small groups such as hunger strikes and self-immolation. Consequently, protest event specific annotation schemas and datasets were proposed [Huang et al., 2016, Sönmez et al., 2016, Makarov et al., 2016]. These kind of resources are mainly created using key terms for a single context and it is a challenge to obtain the datasets on the basis of the shared limited information. We follow the detailed protest event information tradition as proposed by Lorenzini et al. [2016] and Gerner et al. [2002], work on data not restricted with key terms, and make our data available to researchers in terms of shared-tasks and sufficient information to enable any researcher to work on it.

Recent studies often assume one or more of the following i) analyzing thousands or millions of sources will compensate for low tool performance, i.e. recall, ii) a news report contains information about a single event, iii) analyzing a sentence individually is sufficient to extract relevant information about an event, and iv) tool performance on a new source

will be comparable to the performance on the validation setting [Weischedel and Boschee, 2018, Tanev et al., 2008]. Quantifying effect of these assumptions is not a simple task, therefore they are rarely tested. We either provide our observations that shed light on the effect of these assumptions or do not make them.

## 3. Methodology

A gold standard corpus of protest events that can enable large scale, multi-source sociopolitical studies should be representative of the content it aims to capture. Moreover, it should enable quantifying the performance of the automation across contexts. Therefore, using available corpora that are already being allowed to be facilitated in research such as English Gigaword [Parker et al., 2011] is not an option for this setting. In order to satisfy these requirements, our methodology is designed to contain and incorporate multiple sources and countries and apply a detailed annotation methodology without sacrificing quality.

We collect online local news articles, and international sources when local sources are not accessible, from India, China, and South Africa. We first download URLs of the freely accessible parts of online news archives such as Indian Express (IEX), New Indian Express (NIEX), The Hindu (TH), Times of India (ToI) and South China Morning Post (SCMP), People's Daily (PD).[3] Then, for each source, we take a random sample of these URLs and download their content for labelling and annotation.[4]

The random sampling approach makes the task challenging and closer to reality. Key term list causes some events such as the ones reported without using common terms of protest concept (the phrase "classrooms empty" can be used to report on a strike organized by teachers) to be missed [Hürriyetoğlu et al., 2019a]. Moreover, lexical variance across contexts cannot always be captured using key terms. For instance, the terms "bandh" and "idol immersion" are event types that are specific to India and not covered by any general-purpose protest key terms list. Our evaluation of four key term lists, which are reported by Huang et al. [2016], Wang et al. [2016], Weidmann and Rød [2019], and Makarov et al. [2016], yielded .68 and .80 precision and recall on our randomly sampled batches at best.

Our annotation is based on an annotation manual created by an expert, randomly sampling documents from various sources and periods, and continuously monitoring the annotations to achieve a high inter-annotator agreement (IAA). The same manuals are applied on data collected from different sources and countries, which enables obtaining comparable measures of automatic tool performance across contexts. Finally, in order to eliminate the risk of wrong labelling due to lack of knowledge about a country, a domain expert in politics of the target country instructs the annotators before they start the annotation.

The annotation team consists of a supervisor, who is a social scientist and responsible for maintaining the annotation manuals and resolving the disagreements between the annotators, and master students or PhD candidates in social or political sciences, working in pairs. Throughout the annotation, the overlap ratio of annotated articles between pairs is

---

3. The period covered by these archives is between around 2000 and 2017.

4. Only publicly accessible online information is processed and shared in terms of online URLs. We design our data collection, annotation, and tool development in a way that it would not yield any sensitive that could be used to target individuals by malicious state actors information about individuals. The precautions considered are: using and distributing data via URLs, and express personal characteristics in terms of broad categories such as student or worker.

100%. The annotation starts by labelling whether a news article mentions a protest. Then, the sentence(s) that contain protest information are identified. Finally, protest information such as participants, place, and time is detected in the protest-related sentences at token level. The three levels of annotation, are separate but integrated in the sense that they form a pipeline in which a single document goes through each individual step, and each step is built upon the result of the previous step, which is about completing a batch of documents for a specific level before we start to annotate at next level. The aim here is to maximize time and resource efficiency and performance by utilizing the feedback of each level of annotation for the whole process. The lack of clear boundaries between these levels at the beginning of the annotation project had caused a relatively lower IAA and more time to be spent on the quality check and correction of the dataset. For these same reasons, we add a new step, namely sentence level, to the aforementioned main steps of protest event pipelines. This order of tasks enables error analysis and optimization possibility during annotation and tool development efforts.

Each batch for document and sentence levels is corrected in terms of

**Spotchecks** %10 of the agreements were checked by the annotation supervisor.

**ML-internal** 80% of the batch is used to create an ML model. Next, the remaining 20% is predicted using this model. This procedure was repeated until all instances are used at least once in training and once in test data.

**ML-external** The annotated and corrected data from previous batches is used to create an automated classifier that is used to classify the newly annotated batch.

The disagreements between the classifiers and the annotations are checked manually for ML-internal and -external. Annotations that were checked based on spotcheck, ML-internal, and -external were found to be annotated wrongly at around 2%, 50%, and 10% respectively. In total, around 10% of the annotations were corrected using these measures.

In the cases we need to improve the performance of a tool on an already covered source, adapt it to a recent period, or start the analysis of a new source, we apply a recall optimized active learning (AL) based sampling from a news archive. We first train multiple ML-based classifiers (three or more) on the available corpus, then predict a random batch from the new context. To achieve elevated recall scores, we take the logical-or of all classifiers as final prediction, and select positive samples to be annotated. Although the recall decrease from 100% to 97% in such a sample, the precision of it increase from around 5% to around 70% in comparison to a random sample.[5] AL significantly decreases the effort the annotators should spend on annotation [Settles, 2009]. Since annotators observe many more positive samples in this setting, we expect the decrease of recall to be a minor issue at the moment.

## 4. Annotation manuals

Each annotation task has its respective annotation manual which defines the task and lays out the rules of annotation, which also enumerates certain cases which might be confusing

---

5. This performance was measured on an AL sample and 200 news articles that were excluded from annotation at this sampling operation. The training data consisted of around 4,000 news articles that were randomly sampled and annotated from the same country of the resulted AL batch.

and elaborates the context of a number of concrete situations and examples, for that task.[6] The linguistic units of annotation, i.e. types of texts, for consecutive annotation tasks are documents, sentences, and expressions, i.e. groups of tokens respectively.

**EMW Event Labeling Manual (ELAM)** is created for the document level annotation. This manual lays out the protest event ontology, that is, the protest event definition which specifies the range of CP events that are included in the scope of the project. Also, it contains the rules by which the news articles are decided to contain CP events. In a nutshell, CP events cover any politically motivated collective action which lays outside the official mechanisms of political participation associated with formal government institutions of the country in which the said action takes place. This broad event definition is developed and fleshed out in two sections. The first section identifies three abstract categories of collective action, namely, political mobilizations, social protests, and group confrontations, in order to define the broad range of socio-political events that the project simply refers to as protest events. Next, five specific categories of CP events are identified as concrete manifestations of the types of collective action already defined. Demonstrations, industrial actions, group clashes, political violence, armed militancy, and electoral mobilization events are the concrete types of events that the event ontology of the project encompasses. Once the event definition is laid out as such, certain criteria to which the news stories that report protest events must conform in order to be classified as protest news articles are enumerated. These criteria are the necessity of civilian actors, and the existence of concrete time and place information which ascertains that the event(s) the report mentions has definitely taken place. Only the news reports that mention events that have taken place in the past or are taking place at the time of writing are labeled as protest news articles. The references to the future (i.e. planned, threatened, announced or expected) events are not labeled, with the exception of threats of or attempts at violent events.[7]

**EMW Event Sentence Annotation Manual (ESAM)** enumerates rules according to which sentences of news reports are classified into protest event sentences and non-event sentences. Similar to the document level, sentences which contain references to the protest events are labeled as protest event sentences. The protest event sentences are defined as those sentences which give information about an event present in the news report and contain at least one direct reference to a protest event. That is to say, all event sentences must contain an expression which denotes the event.

**EMW Information Extraction Annotation Manual (IEAM)** acts as the guide to the annotation on the token level. The IEAM defines all variables, pieces of information about protest events that the EMW project aims at extracting, and lays out the rules according to which expressions in the event sentences are to be annotated by using tags. There are general rules which apply to all tags, as well as specific rules which apply to individual tags. The tags and their corresponding variables are grouped under event, participant, organizer and target characteristics. Event characteristics are those variables that give information about the event itself, the main one being event trigger, i.e. event expressions

---

6. The annotation manuals can be found on https://github.com/emerging-welfare/general_info/tree/master/annotation-manuals.

7. Although planned events and protest threats could have a role in our analysis [Huang et al., 2016], they are neither relevant in the CP context nor their prevalence, which is below 0.5% of a random sample according to our observations, allow their automated analysis.

which either directly denote the event or refer to it. The rest of the event characteristics and all other variables that are listed in participant, organizer, and target characteristics are event arguments. As it is mentioned above, event triggers are linguistic units which must exist in event sentences. All event arguments that exist within the text will only be annotated in event sentences. This is to ensure that arguments belong to their respective events unambiguously. Semantic categories of events are different types of collective action that the events in the articles fall into. Demonstrations (rallies, marches, sit-ins, slogan shouting, gatherings etc.), industrial actions (strikes, slow-downs, picket lines, gheraos etc.), group clashes (fights, clashes, lynching etc.), armed militancy (attacks, bombings, assassinations etc.) and electoral politics events (election rallies) are the event categories involved. Every event trigger is annotated as event type or mention, as well as one of the semantic categories that it is in. Participant and organizer characteristics contain participant type, name, ideology, religious, ethnic and caste identity, and socioeconomic status information about the actors which engage in protest. Participants are any individuals or groups who actively engage in the protest action, that is they are present at the event itself. Organizers are most commonly organizations that hold or take part in the protest events such as political parties, NGOs, unions etc. In some cases, influential individuals or leaders might be the organizers of the protest events. Individuals are annotated as organizers only in special cases where the article designates them explicitly as organizers or leaders of the protests. Target characteristics consist of target type and name and designates the antagonists of the protest events in the article, should they have one. The protests might target governments, officials, leaders, political organizations or other social groups in case of group clashes.

Below are examples of how event information that is annotated on the token level. The bold tokens are the event triggers. The underlined tokens are event arguments, namely organizer name, participant count, participant type, participant caste, event time, event place, facility type, and target name. Note that the event which has not taken place, the rally, in the second sentence is not annotated.[8]

1. **It** took a communal turn that had resulted in **stone-pelting**, **arson** and **loot**.

2. The Bhim Army and other Dalit groups were refused permission to organize a rally against **atrocities** on May 9, sparking off **violence** and **vandalism**, with several vehicles and buses **burnt**.

3. At noon, BJP workers **gathered** in the square and **shouted slogans**, condemning the failure of the Union Government in delivering justice to the victims of last year's terror **attack** at the train station where armed militants **killed** 25 people.

4. In Bangalore, hundreds of workers **participated** in the **rally** in front of the collectorate.

---

8. We treat the event triggers and any other expressions that have a hyphen between them as a single token, e.g., 'stone-pelting'. But, when there is not a hyphen between words, which is the case for 'shouting slogans', the expression consists of two tokens. The first token is annotated as B-trigger and the following token(s) are annotated as I-trigger.

|             | ES  | INT | IEX | NIEX | PD  | RCV1 | SCMP1 | SCMP2 | TH  | ToI  |
|-------------|-----|-----|-----|------|-----|------|-------|-------|-----|------|
| Protest     | 151 | 262 | 296 | 71   | 69  | 802  | 17    | 19    | 264 | 481  |
| Non-Protest | 149 | 738 | 265 | 630  | 732 | 367  | 985   | 483   | 782 | 1985 |
| Sampling    | K   | AL  | AL  | R    | R   | AL   | R     | R     | AL  | R&AL |

Table 1: Document label statistics and sampling method. K, R, and AL indicate Key term, Random, and Active Learning respectively. ES, INT, and RCV1 are EventStatus, International, and Reuters which is filtered for China using meta-information.

## 5. Corpus

We annotate the corpus in three levels that are i) **Document** is what a reader see on a news article. It consists of a title, publication time, and the article text. ii) **Sentence** is a text unit that ends with a sentence completing punctuation mark, e.g., period or question mark. iii) **Token** is a punctuation mark or sequence of alphanumeric characters that is characterized as word in English. The IAA for document and sentence levels are above .75 and .65 Krippendorf's alpha [Krippendorff et al., 2016] on average. The IAA for the token level is less consistent than other levels. Therefore, it is provided for each information type in Table 3. We interpret these scores as indication of how hard is the task and how the annotation manuals are able to enable the process.

The document counts for each document level batch are reported in Table 1. Each batch is named after the source it was sampled from. In case we annotate some data from a corpus such as EventStatus [Huang et al., 2016] (ES)[9] and RCV1 [Lewis et al., 2004][10] that are readily released, we use their names as batch names. Suffixes were added to distinguish between different batches from the same source. For instance, SCMP1 and SCMP2 differ in terms of the period, which is 2000-2002 and 2000-2017 respectively, they cover.

Active learning was applied to create three batches, which are INT2 (Guardian), SCMP3, and NIEX2 (New Indian Express), of articles that were annotated at the sentence level and reported in Table 2. The whole documents were sampled and their sentences were annotated. The high number of non-protest sentence annotations are caused by the documents that do not contain any protest information.

|             | INT2  | SCMP3 | NIEX2 |
|-------------|-------|-------|-------|
| Protest     | 1,658 | 511   | 1,299 |
| Non-protest | 9,045 | 2,847 | 7,083 |

Table 2: Sentence level statistics. The total number of sentences and their annotations as protest and non-protest are reported.

Sentences of a subset of the positive documents were annotated at the token level. The number of the information types in the annotated documents, which are 704 and 135 from

---

9. https://catalog.ldc.upenn.edu/LDC2017T09, accessed on November 25th, 2019.

10. https://trec.nist.gov/data/reuters/reuters.html, accessed on November 25th, 2019.

| Tag name | Time | Trigger | Place | Facility | Participant | Organizer | Target |
|----------|------|---------|-------|----------|-------------|-----------|--------|
| India | 822 | 1,378 | 645 | 392 | 2,283 | 1,260 | 1,453 |
| China | 144 | 142 | 82 | 52 | 272 | 88 | 109 |
| IAA | 60.07 | 50.02 | 41.82 | 39.10 | 39.50 | 47.44 | 34.38 |

Table 3: Token level statistics and IAA in terms of Krippendorf's alpha

India and China respectively, are reported in the Table 3. A news article is annotated at the token level only if the event happens in the same country of the source or the country under focus for international sources, because protest event characteristics that are different across countries may affect the quality of the annotation.[11]

The separate event count per document is 1, 2, 3, 4, 5, 6 or more in 60%, 23%, 7%, 5%, 2%, and 3% of the documents respectively. Moreover, the distribution of the semantic event types demonstrations, industrial actions, group clashes, political violence and armed militancy, electoral mobilization, and other events is 55.5%, 8.9%, 13.7%, 18.8%, 2%, 1.1%. These numbers illustrate the first quantification of the multi-event and event type distribution phenomena in a random sample of news articles in protest domain and show that the assumption, made for instance by Tanev et al. [2008], of a news article contain a single event inherently misses a significant amount of event information.

The document and sentence level annotations are stored in JSON and token level data is stored in FoLiA [van Gompel and Reynaert, 2013] formats. We distribute the corpus in a way that does not violate copyright of the news sources. This involves only sharing information that is needed to reproduce the corpus from the source in cases it is not allowed to distribute the news articles. Namely, the document and sentence level data is downloaded using software we have developed and packaged in a Docker image. These software tools download, clean, and align text are provided in a Docker image in order to facilitate ease of use and reproducibility. The validation of this software was performed during the aforementioned shared-tasks.[12]

## 6. Evaluation

We have exploited the data from India in the corpus to train ML-based models using BERT [Devlin et al., 2019], for document and sentence classification and token extraction in various scenarios. We fine-tune the pretrained BERT-Base with our data. Of each model for every level, hyperparameters are the same as original authors', except for our sentence classifier which restricts maximum sequence length to 128 instead of 512.[13] Table 4 provides F1-macro scores of the document and sentence classification and F1-score that is based on CoNLL 2003 evaluation script for token extraction models for held-out data from India, for

---

11. Around 10% of the positively annotated documents at the document level in a random sample reports a protest event that does not occur in a country under focus.

12. Please follow the instructions on the Global Contentious Politics Gold Standard Data (GLOCON GOLD) repository to obtain the corpus: https://github.com/emerging-welfare/glocongold

13. For our document model, if we split the text in subparts smaller than 512 tokens and take the logical or of each subpart's prediction as that document's prediction, the performance increases 2-3 F1 macro points in comparison to just using the first 512 tokens in a document for prediction.

China and international[14] data in the corpus. The token extraction score is only the trigger detection performance in this table.

|  | India | China | Int-China | South Africa |
|---|---|---|---|---|
| Document | .89 | .82 | .83 | .85 |
| Sentence | .85 | .79 | .83 | .85 |
| Token | .74 | .67 | N/A | N/A |

Table 4: F1-macro of document and sentence classification and F1 for trigger extraction.

The token level scores, which are based on the BERT-base model fine-tuned on our GSC and are generated using a held-out part of it, are reported in Table 5. Additionally, we fine-tuned the Flair NER model [Akbik et al., 2018][15], which is trained on CoNLL 2003 NER [Tjong Kim Sang and De Meulder, 2003], on our data by mapping our place, participant, and organizer tags to "LOC", "PER", and "ORG" in CoNLL data respectively. This model yielded significantly better results, which are .780, .697, and .652 for the place, participant, and organizer types respectively, in comparison to the BERT-base model. Finally, we run an event extraction model, which is again a BERT-base model, that is trained on ACE event extraction data on the same test data. We measured the trigger detection performance of this model based on its CONFLICT category predictions. The F1 scores of the CONFLICT type are .543 and .479 on its own and on our new data respectively. The difference between the scores obtained using ACE and our training data show that our efforts contributes to the protest event collection studies significantly.

We integrate the tools reported in Table 4 and report their performance on a separate 200 news articles dataset, which consists of 100 positively and 100 negatively predicted documents at document level, from India in Table 6.[16] *Doc*, *Sent*, and *Tok* correspond to the tool applied in the order the tool name is mentioned in the configuration name. The highest precision, recall, and F1-macro was yielded by *Doc+Sent+Tok*, *Tok*, and *Doc+Tok* respectively. The event trigger detection score is the reported one for the *Tok*. The performance of the trigger detection is lower than the one reported in Table 5, since this evaluation setting contain non-protest documents as well. The obvious result is that each additional component improve precision, but decrease the recall. The interesting result here is that integrating only the document classification tool enhance precision and a slight decrease in recall in comparison to other configurations.

---

14. The international data is our Guardian sample that is filtered using active learning for China.

15. https://github.com/flairNLP/flair, accessed on April 5, 2020.

16. We exclude 15 documents that contain events not related to India.

|  | Trigger | Time | Place | Facility | Participant | Organizer | Target |
|---|---|---|---|---|---|---|---|
| Precision | 0.756 | 0.663 | 0.724 | 0.436 | 0.649 | 0.568 | 0.497 |
| Recall | 0.691 | 0.704 | 0.646 | 0.436 | 0.564 | 0.619 | 0.485 |
| F1 | 0.722 | 0.683 | 0.683 | 0.436 | 0.604 | 0.593 | 0.491 |

Table 5: Token level information extraction scores per information type.

|           | Tok   | Sent+Tok | Doc+Tok | Doc+Sent+Tok |
|-----------|-------|----------|---------|--------------|
| Precision | .624  | .696     | .660    | **.701**     |
| Recall    | **.663** | .561  | .647    | .547         |
| F1        | .643  | .621     | **.653** | .614        |

Table 6: Trigger identification performances in various configurations of a pipeline.

Additional results that were yielded using some parts of this corpus can be found in various publications. The results obtained in a shared-task for the cross-context document and sentence classification and token extraction were reported in the overview paper of the ProtestNews Lab [Hürriyetoğlu et al., 2019b], which was held in the scope of Conference and Labs of the Evaluation Forum (CLEF 2019). Moreover, the dataset was facilitated the event sentence coreference identification task, in which the participants developed systems to identify sentences about the same event in the scope of the workshop Automated Extraction of Socio-political Events from News (AESPEN) at Language Resources and Evaluation Conference (LREC 2020) [Hürriyetoğlu et al., 2020]. Participants of the shared tasks reported comparable results to the performance reported here.

## 7. Conclusion and Future Work

We introduced a GSC that enables benchmarking and creation of automated tools for protest information collection across contexts. The methodology we have developed to ensure quality of the corpus, our observations during applying this methodology, and the results obtained using automated tools created using the corpus were reported in detail. The clear performance drop as the test data differ from training data, which is known as domain or covariate shift problem [Storkey and Sugiyama, 2006], shows how it is critical to incorporate cross-context aspects to the corpus and evaluation. Handling each source separately is the solution to improve reliability of the performance scores. Our ML models were created using training data from only a single country. Following steps should incorporate data from multiple contexts at model creation phase [Li et al., 2018, He et al., 2018].

We keep track of what is included and excluded at each level in order to better automatize the task and allow quantification of the recall, which has been missing in this field. Restricting datasets by using key terms or basing a protest knowledge base on a subset of a source due to practical reasons was harming the validity and reliability of the resulted datasets. Starting with a random sample and continuing with recall optimized active learning during the creation of the gold standard corpus ensures training data will improve quality of the final gold standard dataset.

We will extend the GSC with news sources in English, Portuguese, and Spanish and semantic categories such as violent vs. non-violent and urban vs. rural. Moreover, we will handle documents and sentences that contain multiple events. The recent models assume there is a separate event for each event trigger that is identified by the token extractor [Weischedel and Boschee, 2018]. However, our observations directed us to identify and link the triggers that denote the same event [Ruppenhofer et al., 2010, Gabbard et al., 2011]. We will be developing tools for linking the event triggers about the same event in our pipeline [Lu and Ng, 2018].

## Acknowledgements

The authors are funded by the European Research Council (ERC) Starting Grant 714868 awarded to Dr. Erdem Yörük for his project Emerging Welfare.

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
