# OpenReview forum: "Cross-context News Corpus for Protest Events related Knowledge Base Construction"
_AKBC.ws/2020/Conference — AKBC 2020_

### Official Review · AnonReviewer1 · 2020-03-27
**Interesting corpus, with some clarifications needed**

**Rating:** 7
**Confidence:** 4

**Review:**

The paper describes a corpus of news articles annotated for protest events. Overall, this is an interesting corpus with a lot of potential for re-use, however, the paper needs some clarifications. A key contribution of the paper is that the initial candidate document retrieval is not based purely on keyword matching, but rather uses a random sampling and active learning based approach to find relevant documents. This is motivated by the incompleteness of dictionaries for protest events. While this might be true, it would have been good to see an evaluation of this assumption with the current data. It is a bit unclear in the paper, but were the K and AL methods run over the same dataset? What are the datasets for which the document relevance precision & recall are reported on page 8?

I would also like to see a more detailed comparison with more general-purpose event extraction methods. Is there a reason why methodologies such as [1] and [2] cannot be re-applied for protest event extraction?

A small formatting issue: the sub-sections on page 8 need newline breaks in between.

[1] Pustejovsky, James, et al. "Temporal and event information in natural language text." Language resources and evaluation 39.2-3 (2005): 123-164.
[2] Inel, Oana, and Lora Aroyo. "Validation methodology for expert-annotated datasets: Event annotation case study." 2nd Conference on Language, Data and Knowledge (LDK 2019). Schloss Dagstuhl-Leibniz-Zentrum fuer Informatik, 2019.

EDIT:
Thank you for addressing the issues I raised. I have changed the review to "Accept".

---

> ### Author Response · Authors · 2020-04-08
> **Update to the paper and comments on the review**
>
> We appreciate the time you spent reading our paper and your comments that helped us to improve the paper. We have updated the paper in light of your comments.
>
> Question 1:  A key contribution of the paper is that the initial candidate document retrieval is not based purely on keyword matching, but rather uses a random sampling and active learning-based approach to find relevant documents. This is motivated by the incompleteness of dictionaries for protest events. While this might be true, it would have been good to see an evaluation of this assumption with the current data.
>
> Answer 1: We added the information you requested. The relevant fragment in this new version of the paper "Moreover, lexical variance across contexts cannot always be captured using key terms. For instance, the terms \bandh" and \idol immersion" are event types that are specific to India and not covered by any general-purpose protest key terms list. Our evaluation of four key term lists, which are reported by Huang et al. [2016], Wang et al. [2016], Weidmann and Rod [2019], and Makarov et al. [2016], yielded .68 and .80 precision and recall on our randomly sampled batches at best"
>
> Question 2: It is a bit unclear in the paper, but were the K and AL methods run over the same dataset? What are the datasets for which the document relevance precision & recall are reported on page 8?
>
> Answer 2: The K method was evaluated on our random samples. We performed a specific experiment to evaluate the AL method. We exploited our data to date as training data to create various ML-based classifiers, applied the filtering on a random sample using these models, and annotated all included and 200 excluded documents to calculate these scores.
>
> Question 3: I would also like to see a more detailed comparison with more general-purpose event extraction methods. Is there a reason why methodologies such as [1] and [2] cannot be re-applied for protest event extraction?
>
> Answer 3: We have added details about why we do not use ACE. Because ACE has comparable event types that are more relevant to our work than TimeML. Moreover, we started our work before [2] was published. We will be looking at this and check whether we can use the proposed methodology in [2] could improve our corpus.
>
> Question 4: A small formatting issue: the sub-sections on page 8 need newline breaks in between.
>
> Answer 4: If you mean the error correction methods, we will correct these in the camera-ready version. We are very sorry, we have just understood this may be what you intend with your comment. Updating the paper again may harm the consistency of our answers to the other reviewer's questions. Please let us know if you think we misunderstand you.

---

### Official Review · AnonReviewer3 · 2020-03-28
**This paper describing a new formalism for  annotating political related corpus. They provide a dataset annotated with their guideline and introduced a BERT baseline over their corpus**

**Rating:** 6
**Confidence:** 3

**Review:**

This paper provides a detailed guideline for annotating socio-political corpus.
The detailed annotation of documents can be time consuming and expensive. The author in the paper proposed a pipelining framework to start annotations from higher levels and get to more detailed annotation if they exist.
Along with their framework, they have provided the dataset of annotated documents, sentences and tokens showing if the protest-related language exists or not.

The author also outlines the baseline line of the transformer architecture regarding the document level and sentence level classifications.

The paper describes the details very clearly. The language is easy to follow.
So to list the pros will be as follows:
-introduction of a new framework for annotating political documents,
-annotating a large scale corpus
-They baseline results

Although they have provided the baseline results on the document and sentence level classifications, they have not provided the results of them over the token level task. It would have been interesting to see if those results are also promising.

The author has mentioned that they have three levels of annotations (document, sentence, and token ) to save time and not spent time on detailed annotations of negative labels. Can they examine how many samples are labeled negative and how much time (in percent) and money it reduced for annotations?
Some minor comments:
-In Page 2: I think “result” should change to “resulted” in sentence below:
Moreover, the assumptions made in delivering a result dataset are not examined in diverse settings.

-On page 3 : who want to use this resources. —> who want to use these resources.

-In page 4: We design our data collection and annotation and tool development  — > We design our data collection, annotation. and tool development

-Page 6 : As it was mentioned above —> As it is mentioned above

-You are 1 page over limit, but there are some repetition in annotation manual, especially when talking about arguments of an event, you can just say as mentioned above,

-The author has mentioned that they have three level of annotations (document, sentence and token ) to save time and not spent time on detailed annotations of negative labels. Can they examine how many samples are labeled negative and how much time (in percent) and money it reduced for annotations?

---

> ### Author Response · Authors · 2020-04-08
> **Update to the paper and comments on the review**
>
> We appreciate the time you spent reading our paper and your comments that helped us to improve the paper. We have updated the paper in light of your comments. We corrected the language errors, removed repetitions, and fit the paper in the page limit. More specifically, in response to your comments:
>
> Question 1: Although they have provided the baseline results on the document and sentence level classifications, they have not provided the results of them over the token level task. It would have been interesting to see if those results are also promising.
>
> Answer 1: We added the results of the BERT-base on the token level task. They are in the Table 4 now.
>
> Question 2: The author has mentioned that they have three levels of annotations (document, sentence, and token ) to save time and not spent time on detailed annotations of negative labels. Can they examine how many samples are labeled negative and how much time (in percent) and money it reduced for annotations?
>
> Answer 2: We described the gain we obtained using these three levels as follows in the new version of the paper "The aim here is to maximize time and resource efficiency and performance by utilizing the feedback of each level of annotation for the whole process. The lack of clear boundaries between these levels at the beginning of the annotation project had caused a relatively lower IAA and more time to be spent on the quality check and correction of the dataset. For these same reasons, we add a new step, namely sentence level, to the aforementioned main steps of protest event pipelines."

---

### Official Review · AnonReviewer2 · 2020-03-28
**Very carefully designed corpus**

**Rating:** 8
**Confidence:** 3

**Review:**

After looking over authors responses, I've decided to increase my rating of the paper.  The main concern I original had was sufficiently motivating the need for this specific dataset (compared to existing alternatives like ACE).  The authors (in the comments below) have articulated qualitatively how ACE is insufficient, and demonstrated with experiments that generalization from ACE pretraining to this new dataset is poor.

==== EDIT ====

The authors present a corpus of news articles about protest events.  10K Articles are annotated with document level labels, sentence-level labels, and token-level labels.  Coarse-grained labels are Protest/Not, and fine-grained labels are things such as triggers/places/times/people/etc.  800 articles are Protest articles.

This is very detailed work & I think the resource will be useful.  The biggest question here is:  If my focus is to work on protest event extraction, what am I gaining by using this corpus vs existing event-annotated corpora (e.g. ACE) that aren’t necessarily specific to protest events?  I’d like to see experiments of models run on ACE evaluated against this corpus & an analysis to see where the mistakes are coming from, and whether these mistakes are made by those models when trained on this new corpus.

--- Below this are specific questions/concerns ---

Annotation:
Just a minor clarification question.  For the token-level annotations, how did you represent multi-token spans annotated with the same label?  For example, in “stone-pelting”, did you indicate “stone”, “-”, and “pelting” tokens with their own labels or did you somehow additionally indicate that “stone-pelting” is one cohesive unit?

Section 4:
Mild nitpick;  Can you split the 3 annotation instruction sections into subsections w/ headings for easier navigation?

Section 6
It says your classifier restricts to the first 256 tokens in the document.  But your classifier is modified to a maximum of 128 tokens.  Can explain this?
Why is the token extraction evaluation only for the trigger?

Regarding the statement around “These numbers illustrate that the assumption of a news article contain a single event is mistaken”.
It was mentioned earlier that this assumption is being made.  Can be more clear which datasets make this assumption?
Can also explain how your limit to 128 (or 256?) tokens does/doesn’t make sense given multiple events occur per article?

---

> ### Author Response · Authors · 2020-04-08
> **Update to the paper and comments on the review**
>
> We appreciate the time you spent reading our paper and your comments that helped us to improve the paper. We have updated the paper in light of your comments. More specifically, in response to your comments:
>
> Question 1: If my focus is to work on protest event extraction, what am I gaining by using this corpus vs existing event-annotated corpora (e.g. ACE) that aren’t necessarily specific to protest events?  I’d like to see experiments of models run on ACE evaluated against this corpus & analysis to see where the mistakes are coming from, and whether these mistakes are made by those models when trained on this new corpus.
>
> Answer 1: We have added information about why we do not use ACE in the relevant work section. Moreover, we added information about an experiment in which we tested a model trained on ACE data on our corpus. The mismatch between the event definitions causes the ACE based model to yield significantly lower results on our data.
>
> Question 2: For the token-level annotations, how did you represent multi-token spans annotated with the same label?  For example, in “stone-pelting”, did you indicate “stone”, “-”, and “pelting” tokens with their own labels or did you somehow additionally indicate that “stone-pelting” is one cohesive unit?
>
> Answer 2: We have added the footnote 6 that provide details of our setting. As a short answer: "stone-pelting" is treated as a single unit (token).
>
> Question 3: Can you split the 3 annotation instruction sections into subsections w/ headings for easier navigation?
>
> Answer 3: We have formatted the names of the annotation as boldface at the beginning of each related paragraph. We believe this touch made this section more readable than before.
>
> Question 4: It says your classifier restricts to the first 256 tokens in the document.  But your classifier is modified to a maximum of 128 tokens.  Can explain this?
>
> Answer 4: The document classifier and token extractor models use the token length 512, which is the default value for BERT-base model. The 256 remained there from a previous experiment. We removed this part from the paper. The sentence classification exploits only 128 tokens, since each sentence in a document is predicted separately. The length 128 was sufficient for the sentence level.
>
> Question 5: Why is the token extraction evaluation only for the trigger?
> Answer 5:  We added the evaluation results for other information types.
>
> Question 6: Regarding the statement around “These numbers illustrate that the assumption of a news article contain a single event is mistaken”.  It was mentioned earlier that this assumption is being made.  Can be more clear which datasets make this assumption?
>
> Answer 6: We added a reference, which is Tanev et al. (2008), about Europe Media Monitor. This project facilitates only the first sentence of a news article and does not search for any other event in the rest of the article. We have personal contact with this team and we were advised by them about not searching more than one event in a document. Although we have shared our results with them, they think using many (thousands) of sources would compensate for the information they loose in a document. However, if want to have control over sources, we think we should use fewer sources and benefit from a source as much as possible.
>
> Question 7: Can also explain how your limit to 128 (or 256?) tokens does/doesn’t make sense given multiple events occur per article?
> Answer 7: The limit is only for the sentence classification. We classify each sentence separately. Therefore, the length limit does not restrict us. Each sentence is assumed to contain at least one event at this level. We are working on event coreference resolution to link the sentences that are predicted as containing events at the moment. Our annotations contain this information and we are working on reflecting this in our pipeline.

---

> > ### Comment · AnonReviewer2 · 2020-04-10
> > **Seeking more specifics about responses to Q1 and Q5**
> >
> > Thanks for the response!  If possible, could you expand on the responses you've provided above?  It's one matter to say you've updated the paper (thanks for doing that), but it'd be really valuable to actually know the content of that update.
> >
> > For example, regarding Q1, "We have added information about why we do not use ACE in the relevant work section. Moreover, we added information about an experiment in which we tested a model trained on ACE data on our corpus. The mismatch between the event definitions causes the ACE based model to yield significantly lower results on our data."
> >
> > --> I'd like to know what this reason is.  Can you include here and/or post a shortened version?
> > --> What are these results?  Can you give more specifics?
> >
> > For Q5, what are these evaluation results?  Can you be more specific?
> >
> > Responses to Q2-4, 6-7 are great, thanks.

---

> > > ### Author Response · Authors · 2020-04-10
> > > **Details to Q1 and Q5**
> > >
> > > Thanks for your evaluatipn. Please find extended versions of Answer 1 and 5 below.
> > >
> > > Answer 1:
> > >  -- The difference of the event definition can be stated as follows: "Moreover, the event definition of ACE and TAC-KBP does not capture the contentious politics (CP) events. For instance, the ACE definition of the event type DEMONSTRATE, in itself, is too restrictive to be applicable in terms of a broad understanding of CP for two reasons. First, as it seems to limit the scope of this event type to spontaneous (that is unorganized) gatherings of people, it excludes certain actions of political and/or grassroots organizations such as political parties and NGOs. Protest actions of such organizations sometimes do not involve mass participation despite aiming at challenging authorities, raising their political agendas or issuing certain demands. Putting up posters, distributing brochures, holding press declarations in public spaces are examples of such protest events. Secondly, the requirement of mass participation in a public area leaves many protest actions such as on-line mass petitions and boycotts, which are not necessarily tied to specific locations where people actually gather, and actions of individuals or small groups such as hunger strikes and self-immolation."
> > > -- The results are as follows, quoting from the new version of the paper,: "Finally, we run an event extraction model, which is again a BERT-base model, that is trained on ACE event extraction data on the same test data. We measured the trigger detection performance of this model based on its CONFLICT category predictions. The F1 scores of the CONFLICT type are .543 and .479 on its own and on our new data respectively."
> > >
> > > Answer 5:
> > > The performance of a token extractor based on BERT-base for the information types is 0.722 for Event Trigger,  0.683 for Time,  0.683 for Place, 0.436 for Facility, 0.604 for Participant, 0.593 for Organizer, and 0.491 for Target in terms of F1. More details about this result are available in the newly added Table 4 now. Additionally, we fine-tuned the Flair NER model, which is trained on CoNLL 2003 NER data, on our data by mapping our place, participant, and organizer tags to LOC, PER, and ORG in CoNLL data respectively. This model yielded significantly better results, which are .780, .697, and .652 for the place, participant, and organizer types respectively, in comparison to the BERT-base model.

---

> > > > ### Comment · AnonReviewer2 · 2020-04-14
> > > > **Updated review**
> > > >
> > > > Thanks for the detailed response!  I've revised my review above.

---

### Decision · Program_Chairs · 2020-05-01

**Decision:**

Accept

**Comment:**

The paper presents a corpus of 10K news articles about protest events, with document level labels, sentence-level labels, and token-level labels. Coarse-grained labels are Protest/Not, and fine-grained labels are things such as triggers/places/times/people/etc.

All reviewers agree that this paper is interesting and the contributed resource will be useful for the community, hence we propose acceptance. There were some concerns that the authors fully addressed in their response, updating their paper. We recommend authors to take the remaining suggestions into account when preparing the final version.